# Phenological cycles in the Pantanal woody communities: Responses to climate and soil moisture seasonality

**Julia Arieira**[1,2]*, **Karl-L. Schuchmann**[2,3,4], **Arnildo Pott**[5], **Michelle D. Lanssanova**[6], **Ana Silvia O. Tissiani**[2], **Osvaldo Borges Pinto Junior**[6], **Marinêz Isaac Marques**[2,4]

**1** Science Panel for the Amazon, São José dos Campos, São Paulo, Brazil, **2** National Institute for Science and Technology in Wetlands (INCT-INAU), Centro de Pesquisa do Pantanal–CPP, Federal University of Mato Grosso (UFMT), Computational Bioacoustics Research Unit (CO.BRA), Cuiabá, Mato Grosso, Cuiabá, Brazil, **3** Zoological Research Museum A. Koenig (ZFMK), Ornithology, Bonn, Germany, **4** Institute of Biosciences, Postgraduate Program in Zoology, Federal University of Mato Grosso, Cuiabá, Mato Grosso, Brazil, **5** Institute of Biosciences, Federal University of Mato Grosso do Sul (UFMS), Campo Grande, Mato Grosso do Sul, Brazil, **6** University of Cuiabá-UNIC, Cuiabá, Mato Grosso, Brazil

* juarieira@gmail.com

**Data Availability Statement:** All relevant data are within the manuscript and its Supporting Information files.

## Abstract

This study investigated the influences of regional climate and soil moisture conditions on the vegetative and reproductive cycles of seven savanna and forest vegetation communities of the Pantanal Mato-grossense. Circular analysis of the phenological data revealed the occurrence of interspecific synchronism and seasonal responses in vegetative and reproductive activities, with flowering patterns differing the most between communities. Leaf shedding intensity events in communities were closely linked to climatic seasonality. Over half of the individuals were semideciduous or deciduous, with leaf drop intense events predominantly triggered by drier, warmer conditions. The annual flood pulse further constrains woody plants, influencing deciduousness and serving as a strategy to mitigate soil water stress. The preceding climatic signal announcing cold fronts was a relevant determinant of flowering events for many communities. Climate and soil seasonality had limited influences on fruiting phenology across the various vegetation communities. The asynchronous response of phenological cycles to surface soil moisture seasonality highlights the diverse habitat hydrodynamics and its interactions with the plant communities that may decouple leaf fall, flowering, and fruiting from surface soil water availability. The unique phenological response of the Pantanal's woody communities to the hydro-climatic cycle sets it apart from other non-flooded savannas of tropical South America. This response involves the intricate interplay between phenological dynamism and alternating drought and wet-flooded phases.

## Introduction

Tropical ecosystems face unprecedented crises. Accelerated changes in global and regional climate and land cover degradation driven by unsustainable models of development are now

**Funding:** This work was funded by the Brazilian Science Council (CNPq) awarded to KLS and MIM, Brehm Foundation for International Bird Conservation, Germany, awarded to KLS, National Institute for Science and Technology in Wetlands (INAU/ UFMT: www.inau.org.br), Brazil, awarded to KLS and MIM, and Instituto Arapyaú awarded to the Science Panel for the Amazon, Brazil, JA.

**Competing interests:** The authors have declared that no competing interests exist.

pushing natural ecosystems in tropical regions to tipping points with loss of resilience [1]. Compared with South American upland rainforests, floodplains and riparian wetlands are among the most important inland wetlands in terms of the ecosystem services provided to people; vegetation composition and function are directly linked to the services provided, such as nontimber products, erosion control, carbon sequestration, and habitats for species [2–5]. Covering approximately 20% of South America, more than 2 million km$^2$ [6], the extent of natural wetlands continues to decline, and insufficient information on the state and impacts from human activities limits their protection [7,8].

In floodplain wetlands, cyclic biological events in plants, such as leaf fall, flowering, fruiting, and seed germination, are indicators of plant functional strategies and often differ along soil hydrological gradients [9,10]. Flood timing and predictability are influenced by seasonal climate rhythms and are the primary drivers of plant species zonation, assemblages, and seasonality of ecosystem productivity [11–14]. Species adaptations to the flood cycle vary widely: some tree species exhibit dormancy followed by leaf fall during flooding, whereas some evergreen trees tolerate flooding by maintaining their metabolism and leaves [15]. The end of the wet season is known to trigger the production of new leaves and fruits on deciduous plants, whereas for evergreen species, sclerophyllous foliage is an adaptive trait that reduces water loss via transpiration. The synchronicity between seed dispersal and hydrology is another adaptation to the flood cycle, which may increase germination success [12].

Approximately 1,000 species of trees that are tolerant to different flood levels exist in Amazon floodplains [16]. In other extensive South American wetlands subjected to predictable monomodal flood pulses, such as the Pantanal Mato-grossense [17], the number of flood-tolerant trees is much lower: approximately 355 species (47% of the Pantanal trees) [18]. Among the 2,567 known Pantanal flora [19], 756 species are classified as woody plants, representing 39.7% of the higher plants [18]. Unlike in the Amazon, the vegetation of the Pantanal is predominantly savanna, with most trees inhabiting nonfloodable environments and being tolerant to prolonged dry periods [18,20]. The Pantanal has a savanna climate with marked seasonality defined by a dry season for more than 5 months. In such a hyperseasonal savanna, as described by Eiten in 1982 [21], alternating dry and wet seasons might trigger the phenophase of the production of leaves, flowers, and fruits [22]. Savanna formations may undergo a high vapor hydric deficit that is 30% greater than that of forests in the dry season, affecting species water use, while some forests are more impacted by seasonal hydrological variation [23]. The capacity of habitats to store soil moisture during periods of low rainfall might provide humid microclimates to plants sensitive to drought [24]. The evolutionary response of species to such seasonal water deficits may involve deep roots and stomatal or chemical control of water loss through evapotranspiration and leaf shedding [23,25]. As in other tropical wetlands [7,8], knowledge about the reproductive and vegetative cycles of woody plants in the Pantanal and their response to seasonality is insufficient [22], limiting our understanding of the functions of ecosystems and the impacts of human-induced environmental changes on biodiversity and ecosystem services. We do not yet know, for example, the impact of climate change on the Pantanal landscape composition and configuration, particularly under scenarios of 30% rainfall reduction and up 7°C temperature increase by 2100 [26]. Such changes could significantly affect the production of flowers and fruits, which are essential resources for some of the richest herbivore wildlife found in South America [27,28]. The intense wildfires of 2019–2020 in the Pantanal have challenged scientists in understanding the regeneration capability of plant communities, which might depend on the speed and direction of reproductive and vegetative phenological responses at the intra- and interspecific levels mediated by trade-offs in energy allocation for different parts of the plant [29].

In this study, we investigated the existence of seasonal phenological patterns in the distinct woody communities of the northern Pantanal Mato-grossense, which are subjected to similar regional climates but under distinct temporal variabilities in soil moisture availability. To determine the underlying factors influencing plant phenological cycles, we investigated the following question: How does the annual seasonality of meteorological conditions and local soil moisture contribute to orchestrating the synchronicity of reproductive and vegetative phenological activities among coexisting species populations?

## Methods

### Climate and hydrogeomorphologic conditions of the Pantanal basin

The Pantanal basin spans Brazil, Bolivia, and Paraguay in the Southern Hemisphere tropics. It has complex meandering and branching river patterns that flow north to south. The central basin is a lowland (<200 m a.s.l.) formed by Quaternary alluvium and is bordered by the old crystalline Central Brazilian Shield in the east and north, the humid Chaco plains of eastern Bolivia in the west and Paraguay in the south, and the young uplifting Andes in the west [30,31]. Located in the upper Paraguay Basin and extending over 150,502 km$^2$ in the central-western region of Brazil, the climate of the Pantanal is typical of tropical savanna (Aw) according to the Köppen (1948) classification [32], with a marked seasonality between the periods of dry winter and rainy summer, leading to a wet-dry climate, with an annual mean temperature of 26.5°C and rainfall of 1,100 mm ± 300 mm concentrated from October to April (~80%) [20,33]. Rainfall decreases sharply in winter, with monthly records of 10 mm, resulting in a dry season in which evapotranspiration surpasses precipitation [34].

The Pantanal is considered a floodplain with an annual and predictable pattern of flooding influenced by river overflow and summer rainfall [30,35,36], controlling geomorphology and the great diversity of terrestrial, aquatic, and temporary terrestrial habitats [17,31]. The plain has a low-elevation and low-relief topography, with altitudes varying between 80 and 150 m and a north-south declivity of 2.5 – 5.0 cm.km$^{-1}$, which allows the seasonal flooding of the Upper Paraguay River to cover an area greater than 130,000 km$^2$ [31]. The level of rivers in the northern Pantanal, such as the Cuiabá River, follows the trend of seasonal rainfall, causing annual flooding of adjacent plains, which can last more than 6 months [37]. Even after the dry season, part of the plain remains inundated for an additional 2 months due to the low slope and poorly drained soil [38,39].

The soils of the Pantanal are generally hydromorphic, with textures varying from clayey to sandy [40]. Many soils remain water saturated or inundated for a few days to several months due to seasonal floods, restricted drainage, or a highwater levels (2.5 m on average) [37,41]. Thus, hydromorphic soils are found across the region, and their morphological characteristics are attributed primarily to the reduction and oxidation of iron and manganese, which also confer their distinctive color [42]. The soils found in the northeastern Pantanal are predominantly clay in texture (between 3% and 65% of clay) and are clayey and in the great domain of the Plinthosol [41].

Alternating annual and multiannual dry and wet periods have resulted in different patterns of discharge and sediment load across the floodplain, creating a mosaic of geomorphologic formations linked to subtle changes in relief, such as high levees and paleolevees, reflecting inundation regimes and the capacity of the soil to retain water [43]. This mosaic is covered by diverse grassland, savanna and forest vegetation with distinct structural, floristic, and functional identities, which constitute the basic units, namely, macrohabitats, for the management and protection of biodiversity [17].

## Phenological data collection and analyses

**Sampling design and collection.** The study area is located along the margins and flood-plains of the Cuiabá River and its tributary, the Piraim River, within the SESC Baía das Pedras. It belongs to the Cuiabá fluvial fan system, the second largest megafan of the Pantanal Basin [43]. This area is part of the SESC Pantanal Ecological Station, located in the municipality of Poconé (Mato Grosso, Brazil), and it is dedicated to ecotourism and scientific research. Covering 4,200 hectares, it hosts the Advanced Research Base for Pantanal Studies of the Federal University of Mato Grosso.

This study was performed during the annual hydroclimatic cycle from July 2017 to July 2018 in the northern Pantanal (16°30'S and 56°25'W). This period represents normal to moderately wet conditions intercepted with intense rainy periods lasting from nearly 2–4 months [44], considering that the cumulative annual average precipitation in the region was 1,516 mm from 1991–2020 [45]. The vegetation communities of the study area represent mosaics of different forest and savanna formations in the northern Pantanal and other subregions [17].

Observations of phenological aspects were made in seven communities, namely, Seasonally Flooded Woodland, also known as Parkland Savanna (PS); Low Tree and Shrub Savanna (Cerrado Savanna—CS); Low Tree and Shrub Savanna, invaded by Trees (Invaded Savanna—IS); Shrubland (SHB); the Alluvial Seasonal Semideciduous Forest (ASDF); the Lowland Seasonal Semideciduous Forest (LSDF); and the Semi-evergreen Monodominant Forest of *Vochysia divergens* (MF) [46].

**Parkland Savanna** occurs in areas experiencing short-term flooding from rainfall accumulation lasting only a few weeks, with depths reaching less than 50 cm, and is distributed along abandoned lobes on fluvial fans [43,47]. These woodland savannas usually encompass a single tree or shrub species over a grassy layer, featuring typical species such as *Byrsonima cydoniifolia* scrub, *Curatella americana*, *Handroanthus heptaphyllus*, and *T. aurea*, which create spots of monodominant formations. The presence of species such as the spiny *Machaerium hirtum* and the palm *Copernicia alba* gives these formations physiognomic similarities with the Steppic Park Savanna (Humid Chaco) of the southern Pantanal (Silva and Caputo 2010, Polido and Sartori). The treelet *B. cydoniifolia* ("canjiqueiral") is associated with sandy soils, whereas *C. alba* is associated with calcium- or sodium-rich patches [48].

**Cerrado Savanna** (Low Tree and Shrub Savanna) is composed of herbaceous and shrubby plants, palms, and isolated trees up to 8 m tall, including species such as *Cupania vernalis*, *Erythroxylum anguifugum*, *Eugenia florida*, *Genipa americana*, and *Tabebuia aurea*. This vegetation community typically occurs when flooding lasts less than 3 months and when the floodwater is up to 1 m deep. These dynamic habitats feature species with high tolerance to floods and droughts [47]. The reduction in flooding and soil moisture leads to species turnover, with some species, such as the less flood-tolerant *Callisthene fasciculata*, becoming dominant.

**Invaded Savanna** (locally called *campo sujo*) by native trees and shrubs exemplifies the Pantanal vegetation dynamics regulated by multiannual dry and wet periods that result in a reorganization of the spatial distribution of vegetation communities and species [47]. Multiple wet years drive the expansion of pioneer and native riparian tree *V. divergens* over flooded open-canopy savannas and natural grasslands, reducing herbaceous diversity and evolving into a semi-evergreen monodominant forest [49]. In this woody encroached community, flooding lasts 3–6 months in wet years, and the soil can remain waterlogged in the dry season, with establishment benefitting from low wildfire regimes [50].

**Shrublands** are usually characterized by encroachment stages of a set of shrubby species over grasslands and savanna areas, such as the entangling *Combretum lanceolatum* and *C.*

*laxum* ("pombeiral"). Shrublands dominated by *C. lanceolatum* are flooded at approximately 2 m and for up to 6 months and have some associated species, such as *Triplaris gardneriana*, a riparian tree. These shrubs initially grow in isolated clumps but gradually form dense mono-specific stands [49].

**Alluvial Seasonal Semi-deciduous Forests** are located along permanent watercourses and experience periodic flooding for short to long periods depending on their position along the relief gradient [51]. The canopy height can reach 30 m on better-drained riverbanks near running water [46,48,52]. It is among the richest woody species colonized by highly flood-tolerant species, including *Garcinia brasiliensis*, *Mouriri guianensis*, *Ocotea suaveolens*, *Siparuna brasiliensis*, *Trichilia catigua*, *Vitex cymosa*, and *Zygia inaequalis*.

**Lowland Seasonal Semideciduous Forests** may appear contiguous to the Alluvial Seasonal Semideciduous Forests along the relief gradient [52] on flood-free or shortly inundated (1–2 months) ridges or paleolevees [53,54]. Many species, such as *Anadenanthera colubrina*, *Aspidosperma cuspa*, *Astronium fraxinifolium*, *Attalea phalerata*, *Cordia glabrata*, *Dipteryx alata*, *Pseudobombax marginatum*, and *Tabebuia roseoalba* are frequently shared between these types of forests. In these slightly higher-lying areas, differences in water availability may benefit species with greater tolerance to drier conditions, such as *Combretum leprosum* [55].

**Semi-Evergreen Monodominant Forest of *Vochysia divergens*,** known as "Cambarazal", is characterized by the monodominance of *V. divergens* (>50% dominance) which is usually found on young fluvial deposits [52]. The understory is composed of medium to large trees such as *Duroia duckei*, *Leptobalanus parvifolius*, and *Nectandra amazonum*, riparian species phytogeographically linked with Amazon flora [46]. The canopy can exceed 20 m high, creating dense tree cover with a sparse understory where *V. divergens* seedlings that are either scattered or absent. This forest typically develops under a prolonged flood regime, often lasting over six months [47].

In each of these seven communities, one to three plots measuring 5 m x 80 m were used to monitor phenological aspects at the community level, totaling 18 plots and a sampled area of 7.200 m$^2$ (0.72 hectares). In each plot, all woody individuals with a DBH $\geq$ 5 cm and palms were identified and tagged, and phenological data were recorded monthly, summing 983 individuals belonging to 59 species across 28 families (17 individuals per species on average). The plant species were collected and deposited in the Herbarium of the Federal University of Mato Grosso (UFMT). Since the research was conducted outside protected areas and did not involve endangered species, no data collection permission from official environmental institutions (i.e., Instituto Chico Mendes de Conservação da Biodiversidade—ICMBio) was required. We identified plants at the species level by comparison with herbarium specimens and with help from taxonomists. Among the 983 individuals, 156 individuals from 14 species were in the PS, 198 individuals from 21 species were in the CS, 138 individuals from 16 species were monitored in the IS, 46 individuals from 5 species were in the SHB, 133 individuals from 34 species were in the ASDF, 180 individuals from 30 species were in the LSDF, and 132 individuals from 15 species were in the MF. The number of species in our study plots is similar to tree richness estimates from other phytosociological inventories conducted in the Pantanal [49,55–57].

We identified vegetative and reproductive phenological events through direct observation with binoculars to evaluate the presence or absence (qualitative) and intensity (quantitative) aspects of phenological dynamics. The vegetative phenophase is represented by leaf fall when canopy gaps can be observed or when leafless branches are present [58]. Flowering encompasses flower buds and anthesis, and the fruiting phase includes green and ripe fruits. Phenological activity and intensity were measured for each vegetation type to evaluate the synchrony, seasonality, and intensity of phenological events in the species populations. The index of phenological activity was assessed through the frequency of individuals in a certain

phenophase, determined by the presence of flower buds and flowers (flowering), green and ripe fruits (fruiting), and leaf fall equal to or greater than 30%, helping to indicate the beginning and end of the phenophase [58,59]. The intensity of the phenological events was measured via the semiquantitative method proposed by Fournier (1974) [60]. We scored the percentages of leaf shedding in five classes by dividing the phenological phenomena into five categories according to intensity: 0 = absence of phenological event, 1 = presence of the phenological event at a magnitude between 1% and 25%, 2 = presence of the phenological event at a magnitude between 26% and 50%, 3 = presence of the phenological event at a magnitude between 51% and 75%, and 4 = presence of the phenological event at a magnitude between 76% and 100%. The Fournier percentage of intensity was calculated monthly by summing the individual value categories (1 to 4) in each phenophase divided by the total number of individuals found in the community multiplied by four. This analysis allows us to estimate the intensity percentage of the phenophase and quantify the abundance of available plant resources and oscillating seasonal productivity [59,61]. The graphical results produced from the seven communities allowed comparisons between qualitative and semiquantitative phenological metrics within and between vegetation communities [59,62]. We classified all individuals as deciduous, semideciduous, or evergreen on the basis of intervals of annual leaf loss. Individuals were classified as deciduous if they lost more than 90% of their leaves at least once a year. Individuals shedding more than 50% of their leaves and less than 90% of their leaves were classified as semideciduous, and evergreen individuals maintained 50–100% of their crowns throughout the year [63,64].

**Phenological synchrony and seasonality.** We applied circular statistics to evaluate species synchronism and seasonality in phenological activities within the seven studied vegetation types, defining seasonality peaks and annual (in activity once a year) and subannual (in activity more than once a year) seasonal patterns [62,65]. This analysis is based on the Rayleigh test (Z) of the effects of the circular distributions of flowering, fruiting, and leaf fall events on the peak vegetative and reproductive phenologies of plants. Therefore, the distributions of the frequencies of flowering, fruiting, and leaf fall in each phenophase were plotted in circular histograms at monthly intervals (from July 2017 to June 2018), with 365 days of the year corresponding to 360˚ of the circumference and months converted into angles of 30˚. The vector length is related to the concentration coefficient ($r_c$) value, which ranges from 0 to 1. The angle where it is plotted indicates the mean angle that corresponds to the phenophase mean date. On the basis of the results, the mean dates (C) and mean angles (μ) of phenophase occurrence were calculated for the month with the highest phenological activity, the concentration of each event around that date ($r_c$; vector length) and the significance value for the seasonal pattern (μ) of the Rayleigh test (p). The C value can be used to determine the seasonality and frequency distributions of occurrences of each phenophase, and the highest concentration represents the highest seasonality [58,61]. Circular histograms of the frequency of leaf fall, flowering, and fruiting events were generated, and the Watson-Williams test was used to test whether the mean periods of event intensity differed among the seven communities [65,66]. The null hypothesis is that means are equal across groups. These analyses were conducted via the software Oriana [67].

**Abiotic control of phenological events.** Assuming that the variables did not have a normal distribution [65], nonparametric Spearman correlations were used to evaluate the relationships between the Fournier intensity data for leaf fall, flowering, and fruiting (response variables) and the regional climate and local soil moisture conditions (explanatory variables). This analysis allows us not to evaluate interactions between variables but rather to contribute to understanding only the most likely phenological triggers, as aimed here [68]. Six regional climate variables were chosen considering their reported influence on phenological dynamics

in tropical regions [69,70], namely, monthly average temperature and mean maximum and minimum temperatures (˚C), actual evapotranspiration (mm month⁻¹), total rainfall (mm), and insolation (hours month⁻¹ of incident solar radiation). The climatic data were acquired from the National Agency of Waters on the website 'Hidroweb' (http://hidroweb.ana.gov.br) [71], and BDMEP–INMET data, representing the station of Cuiabá, Mato Grosso (OMM: 83361). This official weather station has operated since 1911 [72] serving as a reference for climate monitoring in the North Pantanal.

The correlations were conducted with climatic values from the same month ($r_0$) as the date of the phenological data and with values from the first ($r_1$) and second ($r_2$) months before the observations. This approach allowed us to evaluate delays in phenological response to annual climate variation. The local effect of seasonal variation on soil moisture (%) within each community was also analyzed at a depth of 20 cm, a layer where savanna first lateral root branches grow and where soil nutrients are more available [73]. For this purpose, three soil samples per community were collected with a Dutch corer at 20 cm and taken to the Soil Physics Laboratory of the Federal University of Mato Grosso (UFMT) for weighing and drying. The samples were oven-dried at 105˚C for 24 h, and the mean of three samples per unit was considered. The gravimetric soil moisture (%) was determined by the difference between the wet weight (WW) and dry weight (DW) via the equation [(WW–DW)/DW]. A soil moisture content of 100% indicates that the site is in a flooded state. The analyses were performed with the R version 3.6.1 library Hmisc package.

## Results

### Annual variability in meteorological and soil moisture data

The annual air temperature averaged 27.8˚C, reaching a minimum of 15.0˚C in July and a maximum of 39.5˚C in September (Fig 1). Notably, the dry season, marked by low precipitation in July and September, contrasted sharply with the wet season, which accounted for approximately 85% of the total annual rainfall of 1123 mm. These months correspond closely to the annual variations in the water level of the Cuiabá River, the primary catalyst for flooding in the studied floodplain. Evapotranspiration reached its zenith during the rainy season (October – April), peaking at 188 mm month⁻¹. Its alignment with rainfall pattern was inconsistent, suggesting the possible influence of free water surfaces on evapotranspiration during flooding or the impact of minimum temperature fluctuations. The period of intense insolation spanned from July (257 hours month⁻¹) to September (220 hours month⁻¹), coinciding with the lowest actual monthly evapotranspiration (9–38 mm month⁻¹). Conversely, the rainy season corresponded to a 174.3 hour month⁻¹ reduction in insolation in October, which was indicative of increased cloud cover.

The soil water moisture content varied between community sites, as illustrated in Fig 2, depicting the intrinsic connection between surface water flux and vegetation cover, soil drainage capability, and the cumulative effect of rainfall. The lowest average soil moisture content was found in PS (20%±24) and LSDF (20%±4), followed by CS (29%±25), ASDF (30%±19), MF (40%±39), SHR (40%±42), and IS (51%±38) (S1 Appendix). In some vegetation communities, such as the ASDF, MF, and IS communities, increased precipitation was correlated with increased soil surface moisture and greater evapotranspiration in the previous month. There was no apparent direct influence of climate variables on the annual variation in soil moisture in the LSDF and CS. A higher maximum temperature with a peak in October reduced the soil moisture in the SHB.

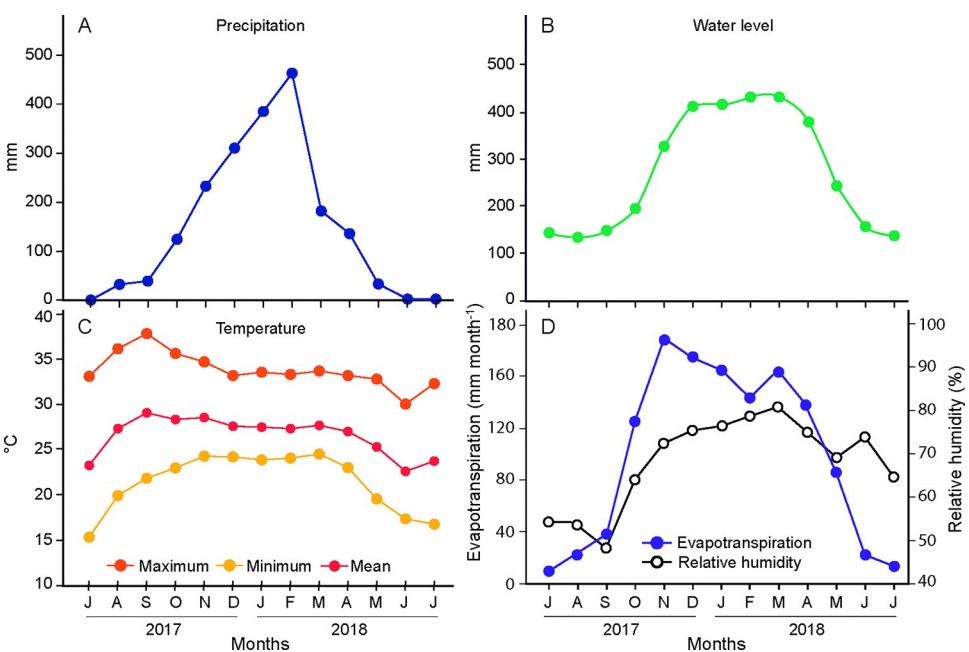

**Fig 1. Hydroclimatic annual variability between July 2017 to July 2018 representing the Cuiabá fluvial fan system.**
A. Precipitation (mm); B. Water level of the Cuiabá River (mm); C. Maximum, mean, and minimum temperatures (˚C); D. Actual Evapotranspiration (mm month$^{-1}$), and relative humidity (%) (sources: National Water Agency (ANA) and the National Institute of Meteorology (INMET)).

## Plant community phenology

**Leaf phenology.** All seven communities had more than 50% of individuals semideciduous or deciduous, i.e., shedding more than 50% of their leaves at least once a year (Fig 3). The communities with a greater number of deciduous individuals were found at both extremes of the soil hydrological gradient. Specifically, IS and MF, located at the wetter end, represented 54% and 23% of deciduous individuals, respectively, whereas PS, which occupied the drier end, represented 51% of deciduous plants. Cerrado savanna accounted for 13% of the deciduous area and 58% of the semideciduous area. The two semideciduous forests contained approximately 50% semideciduous individuals and 10% deciduous individuals, and presented the greatest number of evergreen individuals (LSDF = 39%, ASDF = 35%).

Circular analysis of the phenological data revealed interspecific synchronism and seasonal responses to leaf fall in four of the seven communities, namely, the PS ($r_c$ = 0.21, p< 0.001), ASDF ($r_c$ = 0.19, p< 0.001), SHB ($r_c$ = 0.18, p< 0.001), and MF ($r_c$ = 0.12, p< 0.001) communities (Fig 4A). Most of these communities are characterized by the dominance of a few species. On the other hand, sub-annual seasonal patterns of leaf fall were observed in the activity of the populations in other communities and, to a greater extent, in the intensity of the event, reflected in the low values of the concentration coefficient ($r_c$) for these and other communities.

The average length of the vector represented the average number of consecutive months with the highest population activity. The average month with a high concentration of active species in the PS and ASDF was June, which represented the period of reduced rainfall and a decrease in the moisture content of the local soil. During this period, 94% of the PS species, including *H. heptaphyllus*, *C. fasciculata*, and *M. hirtum*, and 75% of the individuals exhibited leaf loss. In the ASDF, *Cordia glabrata* and *Pseudobombax marginatum* were among the 54%

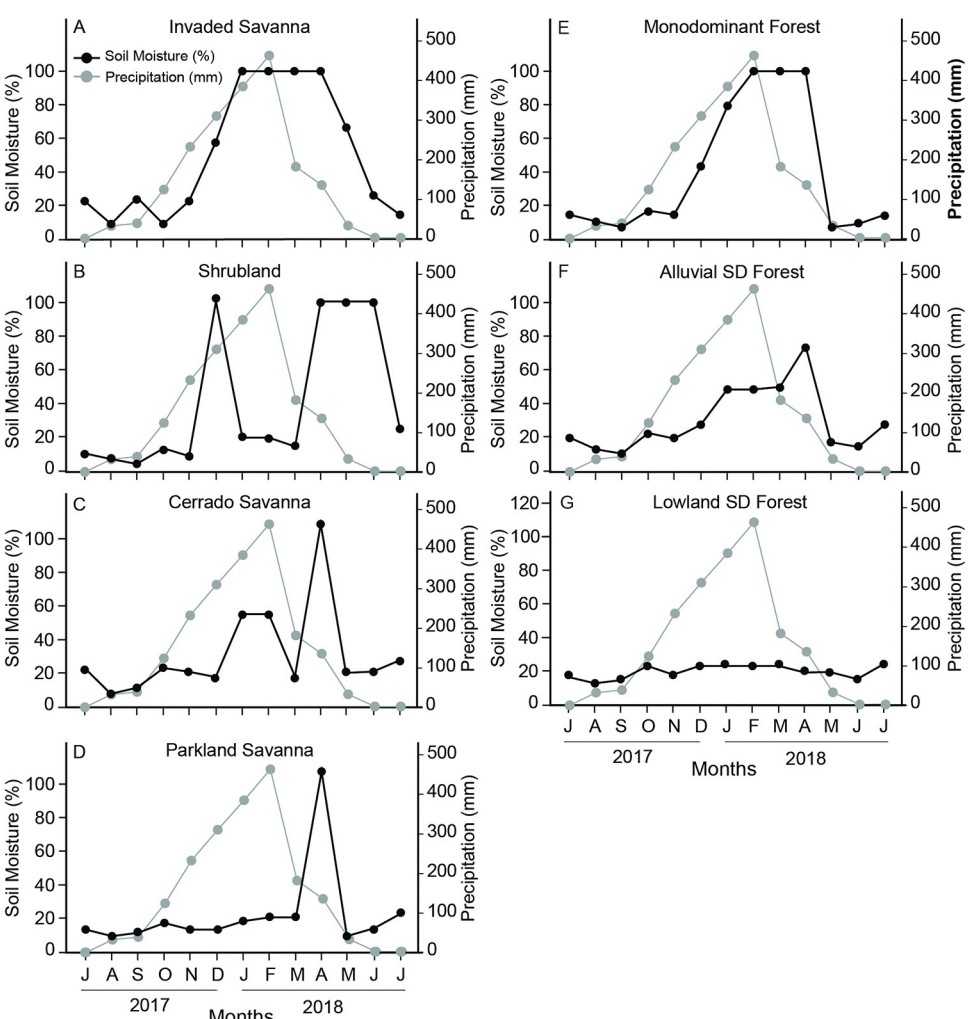

**Fig 2.** Soil moisture content fluctuation (%) in the upper 20 cm of soil across seven plant communities (A–D), alongside precipitation (mm), representing the hydroclimatic cycle from July 2017 to July 2018.

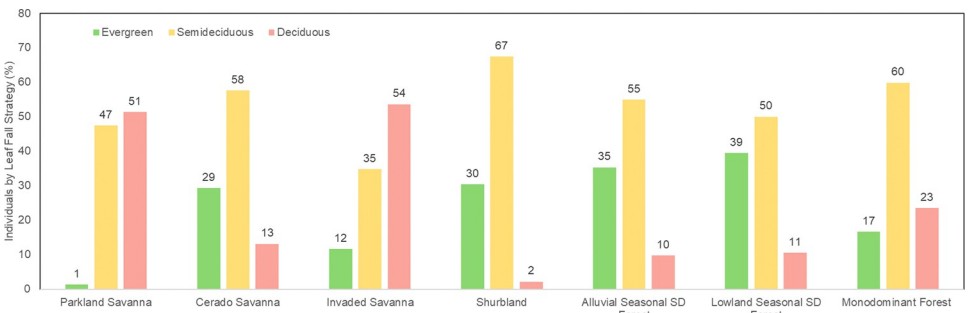

**Fig 3. Individuals by leaf fall strategy (%).** Individuals were classified as deciduous, semideciduous, or evergreen on the basis of intervals of leaf loss. Individuals were classified as deciduous if they lost more than 90% of their leaves at least once a year. Individuals shedding more than 50% of their leaves and less than 90% of their leaves were classified as semideciduous, and evergreen individuals were those which maintained 50–100% of their crown throughout the year.

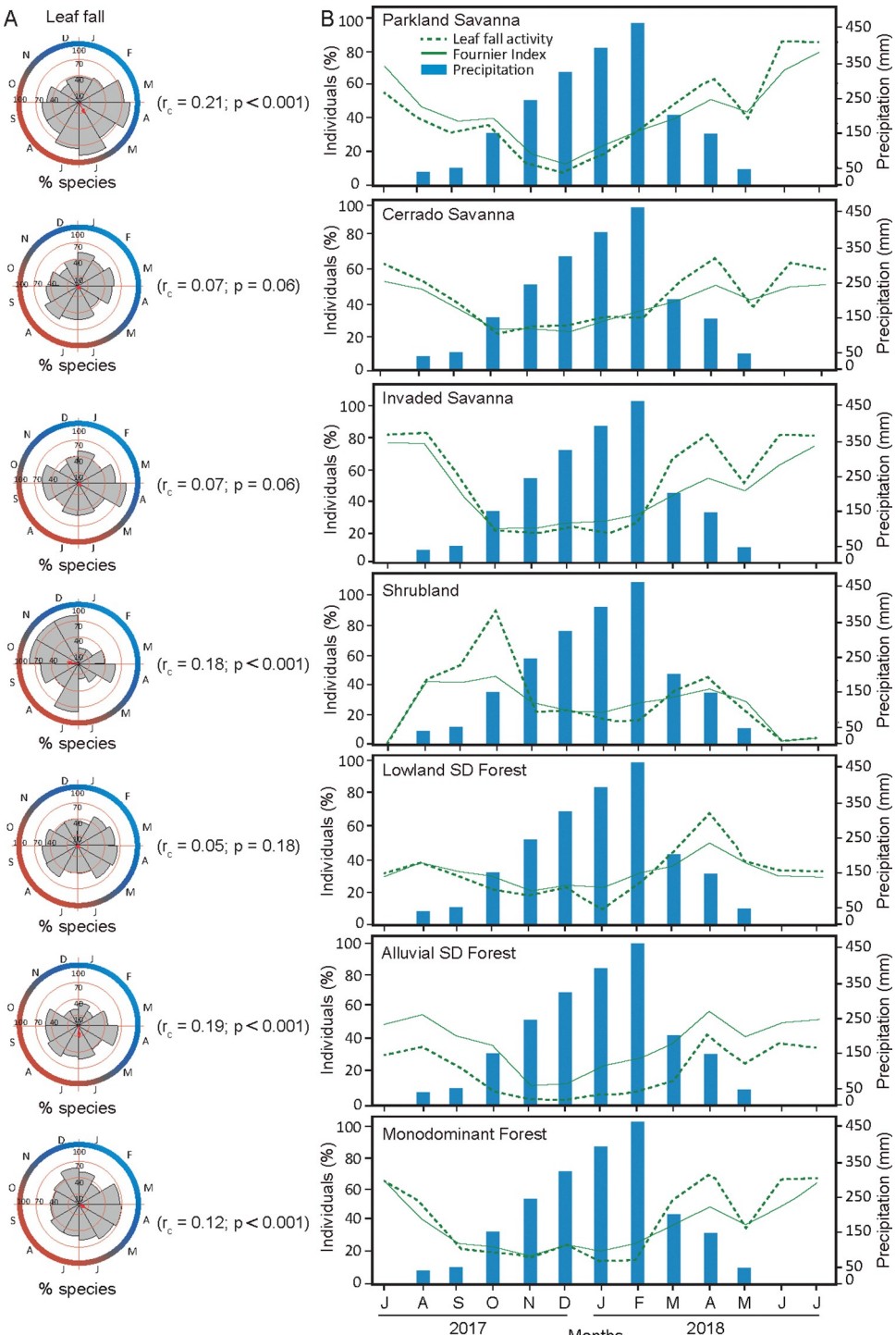

**Fig 4. Vegetative phenology across seven plant communities.** A. Circular histograms of the frequency of leaf fall phenological activity among species. The vector length is related to the concentration coefficient ($r_c$) value, 0 to 1, and the significance value for the seasonal pattern (μ) of the Rayleigh test (p); B. Phenology activity and Fournier intensity of leaf shedding over an annual hydroclimatic cycle (July/2017 to July/2018).

of the species and 36% of the individuals who exhibited leaf-shedding activity. Unlike in the previous communities, in the SHB community, the average month with a high concentration of species exhibiting defoliation activity was October, the transition period between the dry and rainy seasons. During this period, 80% of the species and 96% of the individuals in this community were active, and the event intensity was 49%. The peak community activity coincided with the greatest leaf fall of the pioneer and dominant species *C. lanceolatum*.

MF presented increased leaf fall activity during the rainy and flooded seasons on the floodplain, with the mean month having a greater concentration of species occurring in April. In that month, the frequencies of active individuals (Fig 4B) and species were 68% and 73%, respectively, as observed for the understory species *M. guianensis*, *L. parvifolius*, *Triplaris americana*, *E. anguifugum*, and *Alchornea discolor*. The Fournier intensity (Fig 4B) for leaf fall did not follow the same pattern as did the activity, with the dry season (July, 62%) being more intense than the wet period (49%), which was attributed mainly to the activity of the dominant species *V. divergens*.

IS ($r_c$ = 0.07, p = 0.06) and CS ($r_c$ = 0.07, p = 0.056) presented weak seasonality in terms of leaf fall activity among the species, indicating that fluctuating concentrations of species lost their leaves throughout the year. For the IS, the leaf fall peak occurred during the wet period in April, with 72% of the species being active, such as *E. anguifugum*, *T. gardneriana*, and *V. divergens*. The Fournier intensity revealed intensity peaks during both the wet-flooded and dry phases. Similarly, CS experienced continuous leaf fall activity year-round, with high concentrations of active species occurring in the wet-flooded phase in March and another maximum in the dry phase in August, when 59% of the species were active. The synchronous decrease in leaf fall among individuals was sharper than that between species, with lower percentages of activity occurring during the rainy season (Oct–Feb). The frequency of active individuals increased from 22% in October to 67% in April, when the area flooded. The seasonality of leaf fall in the CS differed from the patterns found in the SHB and LSDF (Table 1). The low soil moisture variation (20%±4%) throughout the year found in the LSDF is reflected in relatively constant leaf fall throughout the year, occurring at levels less than 40% of individuals. The exception occurred in April, when 62% of individuals and 62% of the species were active at a mean Fournier intensity of 47%.

The variation in the proportion of individuals who experienced leaf fall throughout the year generally followed its intensity (Fig 4B) but was not within the same magnitude and had some delays. The relationship between intrapopulation activity and intensity seems to be influenced by the species diversity (or species dominance) found in each vegetation type. The most diverse communities, such as the LSDF and ASDF (30 and 34 species, respectively), had relatively lower proportions of individuals in leaf fall during the dry season (33% and 34%, respectively, in July). In contrast, the less diverse communities, such as the PS (14 species), MF (15 species), and IS (16 species) communities, had high proportions of individuals with leaf fall activity in the same season (86%, 66%, and 83%, respectively).

### Reproductive phenology

*Flowering.* The species' flowering was seasonal for six out of the seven communities (excluding the ASDF community). The highest interspecific synchrony in the flowering period occurred in SHB ($r_c$ = 0.98, p = 0.001) and MF ($r_c$ = 0.83, p = 0.001) when the concentration of plants was high in August-September and June, respectively (Fig 5A). The flowering activity in the SHB was concentrated in months with low precipitation (<40 mm) and soil moisture (<8%). During this low rainfall period, 20% of the species flowered, although few individuals were active (2.17% in August) (Fig 5B). Among the seven communities, the MF presented the

**Table 1. Comparison of mean periods of vegetative and reproductive phenological event intensity across the seven communities via the Watson-Williams test.** The null hypothesis is that means are equal across groups. Significant differences ($p < 0.05$) between community phenological patterns are highlighted with asterisks.

| | Leaf fall | | Flowering | | Fruiting | |
|---|---|---|---|---|---|---|
| **Community** | **F** | **p** | **F** | **p** | **F** | **p** |
| Invaded Savanna x Shrubland | 3.46 | 0.08 | 2.63 | 0.12 | 1.35 | 0.26 |
| Invaded Savanna x Cerrado Savanna | 0.39 | 0.54 | 7.19 | **0.01*** | 0.48 | 0.50 |
| Invaded Savanna x Parkland Savanna | 0.10 | 0.75 | 4.26 | **0.05*** | 0.02 | 0.89 |
| Invaded Savanna x Lowland SD Forest | 3.12 | 0.09 | 5.49 | **0.03*** | 0.70 | 0.41 |
| Invaded Savanna x Alluvial SD Forest | 0.65 | 0.43 | 5.83 | **0.02*** | 0.10 | 0.75 |
| Invaded Savanna x Monodominant Forest | 0.65 | 0.43 | 1.00 | 0.33 | 0.36 | 0.56 |
| Shrubland x Invaded Savanna | 3.95 | 0.06 | 1.23 | 0.28 | 1.04 | 0.32 |
| Shrubland x Parkland Savanna | 2.71 | 0.11 | 0.06 | 0.81 | 0.17 | 0.26 |
| Shrubland x Lowland SD Forest | 0.60 | 0.45 | 0.44 | 0.52 | 1.56 | 0.22 |
| Shrubland x Alluvial SD Forest | 3.19 | 0.09 | 0.56 | 0.46 | 1.44 | 0.24 |
| Shrubland x Monodominant Forest | 2.39 | 0.14 | 3.21 | 0.09 | 1.06 | 0.31 |
| Cerrado Savanna x Parkland Savanna | 0.06 | 0.81 | 5.85 | **0.02*** | 0.37 | 0.55 |
| Cerrado Savanna x Lowland SD Forest | 4.35 | **0.05*** | 1.05 | 0.32 | 1.48 | 0.24 |
| Cerrado Savanna x Alluvial SD Forest | 0.09 | 0.76 | 0.90 | 0.35 | 0.82 | 0.38 |
| Cerrado Savanna x Monodominant Forest | 0.12 | 0.73 | 4.61 | **0.04*** | 0.002 | 0.96 |
| Parkland Savanna x Lowland SD Forest | 2.23 | 0.15 | 0.79 | 0.38 | 1.00 | 0.33 |
| Parkland Savanna x Alluvial SD Forest | 0.20 | 0.66 | 1.28 | 0.27 | 0.21 | 0.65 |
| Parkland Savanna x Monodominant Forest | 0.22 | 0.65 | 3.66 | 0.07 | 0.26 | 0.61 |
| Lowland SD Forest x Alluvial SD Forest | 3.17 | 0.09 | 0.03 | 0.88 | 0.26 | 0.62 |
| Lowland SD Forest x Monodominant Forest | 1.88 | 0.18 | 4.12 | **0.05*** | 1.17 | 0.29 |
| Alluvial SD Forest x Monodominant Forest | 0.01 | 0.94 | 4.22 | **0.05*** | 0.64 | 0.44 |

highest Fournier intensity throughout the annual cycle, with the intensity of phenological events reaching 12% in July. The wet-dry transition season (June to July) marked the reproductive phase in this community, when 36% and 32% of the species (e.g., *L. parvifolius*, *M. guianensis*, and *V. divergens*) and c. 15% of the individuals were active.

The flowering of IS and CS peaked during the dry season, with significant concentrations of species occurring around the mean date in July ($r_c = 0.68$, $p = 0.001$) and September ($r_c = 0.28$, $p = 0.02$), respectively. The major flowering event in the IS occurred from June to September, during which 20–36% of the species and 6–9% of the individuals exhibited activity at an intensity of 3–5%. For example, *T. aurea*, *V. cymosa*, and *V. divergens* flowered during this period.

In CS, despite a weak seasonal pattern of synchrony of flowering events revealed by the low length of the mean vector of the circular analysis (<0.50), the mean date of flower production was significant. There were two peaks of maximum activity in the wet (January, 17% species, 2% individuals) and dry (August, 17% species, 4% individuals) seasons. The Fournier intensity during flowering was low throughout the year, reaching a relatively high value in August (<2%) when the soil moisture was the lowest (8%). Typical Cerrado species, e.g., *E. anguifugum*, *E. florida*, *C. fasciculata*, *C. americana*, and *T. aurea*, were responsible for the highest indices of activity and intensity of this phenophase.

In the LSDF, the circular analysis revealed strong seasonality and interspecific synchrony ($r_c = 0.58$, $p = 0.002$), with a mean flowering concentration in the rainy season of December. The flowering activity peaked in the rainy month of January, when 10% of the species,

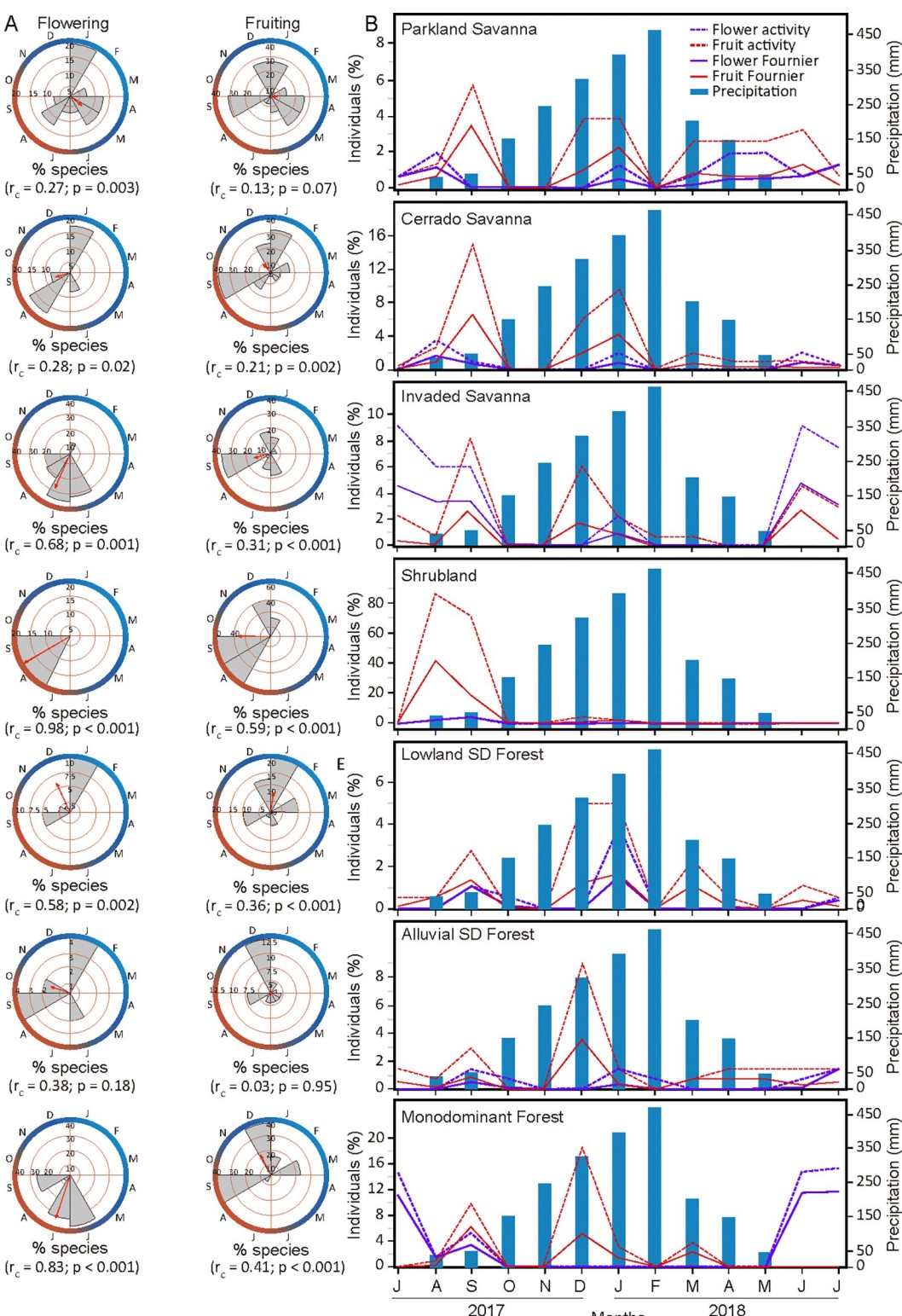

**Fig 5. Reproductive phenology across seven plant communities.** A. Circular histograms of the frequency of species with respect to flowering and fruiting phenological activity. The vector length is related to the concentration coefficient value ($r_c$), 0–1, and the significance value for the seasonal pattern (μ) of the Rayleigh test (p). B. Phenology activity and Fournier intensity of flowering and fruiting over an annual hydroclimatic cycle (July/2017 –July/2018).

including *G. americana*, *T. aurea*, and *T. catigua*, and 4% of the individuals presented flower or floral buds at an intensity of 1.5%.

The PS community presented significant and low interspecific synchrony and a seasonal flowering pattern ($r_c$ = 0.27; p = 0.003), with a mean month of concentration in May. This result highlights the constant, although oscillating, activity in flowering in the eighth of the 12 months of the hydrological year. The months with the highest flowering intensity in the dry season (July-August, 1.1–1.3%) did not coincide with the relatively high proportion of active species and individuals in the rainy season (January, 18% and 1.3%, respectively). A reduction in moisture and rainfall from May to August in PS resulted in an increase in flowering intensity from 0.5% to 1.13%, which coincided with the blooming of *H. heptaphyllus*, a typical species of this community.

In the ASDF, flowering episodes were not synchronous among species ($r_c$ = 0.38; p = 0.18), and the percentages of species and flowered individuals were low throughout the year. We recorded two flowering peaks, one in September, the driest month, and the other in wet January; in both months, the frequency of flowered species was 4%. The annual intraspecific variation in flowering followed the interspecific variation, with January and September corresponding to the months with the maximum frequency of flowering individuals (1.5%). According to the Fournier intensity, the flowering intensity was the highest (0.6%) during dry September when the soil moisture was the lowest (10%), indicating the activity of *M. guianensis* and *T. americana*.

Flowering seasonality was the phenological event with the greatest difference between communities, as indicated by Watson-Williams F tests (Table 1). The IS showed a similar pattern to that found for MF and SHB, communities characterized by a prolonged flood period and monodominance, with flowering concentrated in the dry season.

*Fruiting.* Except for the PS and ASDF communities, all the other five communities exhibited interspecific synchrony during the fruiting cycle. According to the Watson-Williams test, seasonality during the fruiting period did not significantly differ among the communities (Table 1). Fruits were observed in the communities throughout the year, which may be attributed to extended fruiting schedules and staggered ripening times.

SHB ($r_c$ = 0.59, p<0.001) and MF ($r_c$ = 0.41, p<0.001) had the strongest seasonal patterns of interspecific synchrony of fruiting among the communities, with the mean months of fruiting occurring in the dry (September) and wet (December) seasons, respectively (Fig 5A). The few recorded individuals flowering in SHR, followed by high-frequency fruiting, may be due to our monthly monitoring missing the preceding flowering events. In the MF, the frequency of species fruiting was as high in wet December as in dry September (36%), with the event being 17% more intense in September (Fig 5B). Species such as *M. guianensis* and *L. parvifolius* presented fruits during the wet and dry seasons, and others had fruits only in the wet phase, such as *A. discolor*, *Myrcia tomentosa*, and *N. amazonum*. The relatively high percentage of individuals fruiting (18%) coincided with the rainy season in December, approximately three to five months after the flowering peak. This pattern was observed in the monodominant species *V. divergens* (S1 Table).

The mean fruiting date in the IS was dry September ($r_c$ = 0.31, p<0.001), when 36% of the species presented fruits, and the Fournier fruiting intensity was 2.8%. From December to March, during the rainy and flooded periods, fruiting was observed in *E. anguifugum*, *G. americana*, *H. heptaphyllus*, and *V. divergens*.

In the CS community, interspecific synchronism among species ($r_c$ = 0.21, p = 0.002) was associated with the mean concentration of fruiting in the rainy season (November). On the other hand, two fruiting peaks were recorded in CS (Fig 5B), one during the dry season in September (15.3%), when the species *C. fasciculata*, *C. americana*, *Psidium guineense*, and *T. aurea*

were fruiting, and the other during the period of increasing soil moisture in January (9.7%), with fruits observed in *E. florida* and *G. americana*.

In the flood-free LSDF, the Rayleigh test ($r_c$ = 0.36, p<0.001) indicated that the mean date of the fruiting phenophase was in January (Fig 5A), which coincided with the rainy season and the period was the highest soil moisture in this community (49%, Fig 2). The slight difference between the two peaks of Fournier intensity during rainy January (1.7%) and dry September (1.4%) (Fig 5B) did not reflect the increased proportion of active species and individuals. In contrast, the frequencies of active species (20%) (Fig 5A) and individuals (5%) in January were nearly double those reported in September (9.8% and 2.8%, respectively). *Anadenanthera colubrina*, *Attalea phalerata*, *Myrcia tomentosa*, and *Tabebuia roseoalba* were among the fruiting species.

The weak seasonality in fruiting patterns in the ASDF and PS groups suggests that fruiting events were not significantly concentrated at any particular time of the year (Fig 5A). However, in the ASDF, December was the month with the highest relative activity among the species (11%) and individuals (9%) (Fig 5B). Additionally, with the highest fruiting intensity occurring in December (9%), the importance of the rainy season for fruiting in this alluvial forest is evident. The species that entered the phenophase in December were *D. duckei*, *E. anguifugum*, *E. florida*, *M. guianensis*, *O. suaveolens*, and *V. cymosa*. A unique feature of the fruiting cycle in PS is the relatively high concentration of species' fruiting activity during April and May (Fig 5A), marking the abrupt transition from a phase with soil moisture below 20% (March) to an inundated phase (April) (Fig 2). During these months, *M. hirtum* and *Byrsonima cydoniifolia* were observed fruiting. Many species populations in this community, including *M. hirtum* and *H. heptaphyllus*, begin fruiting during the low precipitation season from July to September and continue throughout the rainy season.

## Influence of climate and soil moisture on Fournier's percent index of intensity

**Leaf fall.** The intensity of leaf fall events was correlated with the meteorological variables in all the communities except LSDF (Fig 6). Among all the variables, evapotranspiration had the strongest correlation with leaf fall. The trends of decreasing evapotranspiration, precipitation, and temperatures, coupled with increasing insolation toward the austral dry winter, triggered leaf fall in most of the observed communities. A decrease in the mean (ranging from 25°C to 23°C) and minimum (ranging from 20°C to 17°C) temperatures was associated with increasing Fournier intensity of leaf shedding for most communities (Fig 6B). The negative influence of temperature on leaf drop occurred within two months before the date of maximum leaf fall, indicating that cold fronts enter the region during austral winter. The decrease in minimum temperature from April (23°C) to July (17°C), during the period when rainfall is decreasing and flooding is receding, was correlated with an increase in leaf fall in the PS ($r_0$ = -0.85, p = 0.0002), IS ($r_0$ = -0.81, p = 0.001), MF ($r_0$ = - 0.73, p = 0.005), and ASDF ($r_0$ = -0.71, p = 0.01). Even for other communities with weak species synchrony in terms of leaf activity, such as the CS communities, decreasing trends in minimum temperature were related to increased intensity of leaf fall ($r_0$ = -0.80, p = 0.001). The IS ($r_1$ = -0.60, p = 0.03) and PS ($r_1$ = -0.60, p = 0.03) were also influenced by lower minimum temperatures one month before the phenological event. The mean ($r_0$ = 0.62, p = 0.02) and maximum ($r_0$ = 0.75, p = 0.003) temperatures were the only climatic variables influencing the intensity of leaf fall in SHB. The months with maximum temperatures from August to October (36 – 38°C) corresponded with the maximum leaf fall intensity (43 – 50%).

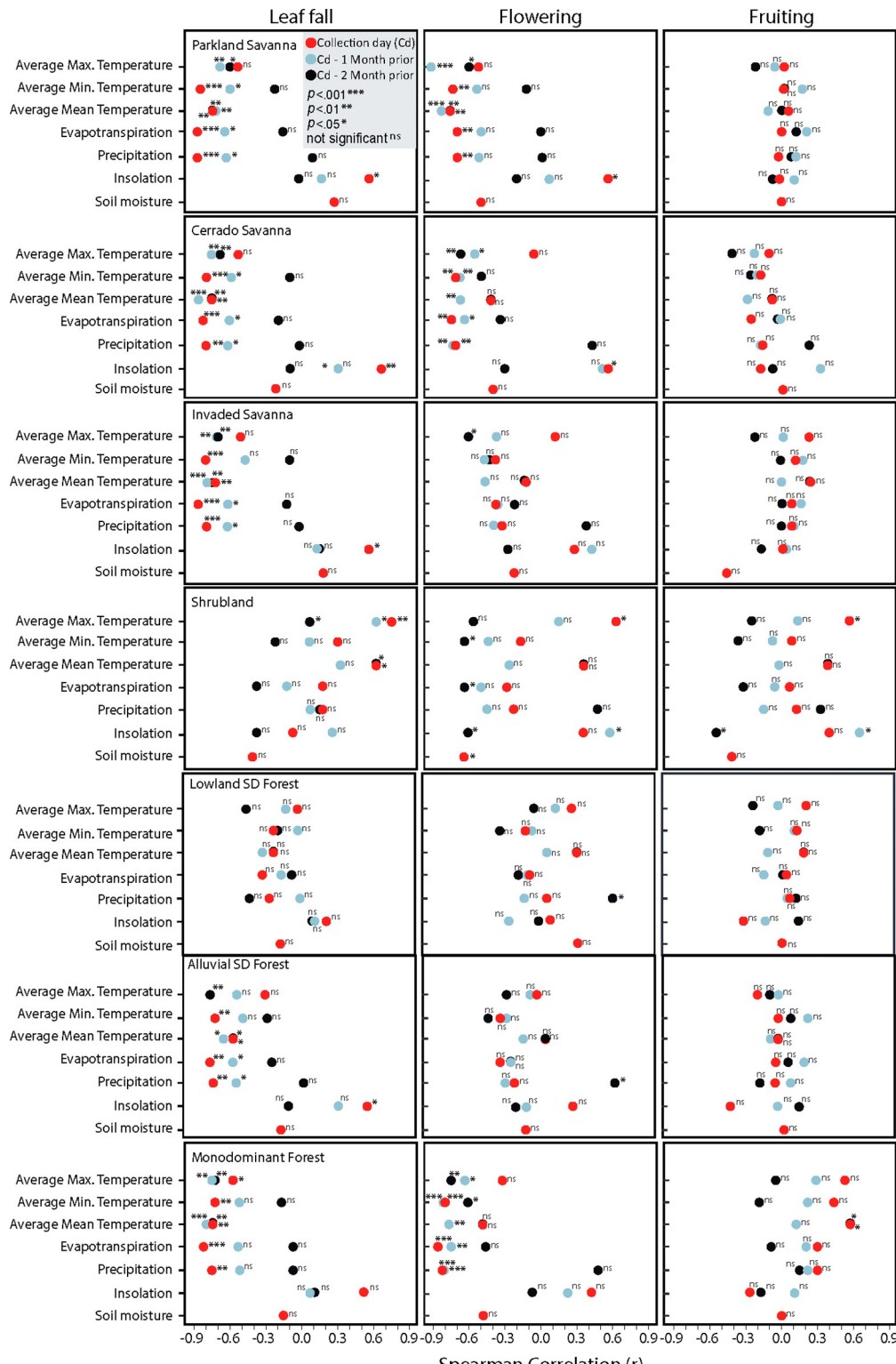

**Fig 6. Spearman correlation analysis between the Fournier intensity data for reproductive and vegetative phenological events and meteorological and soil moisture variables.** The analyses were run considering concomitant (in red), one-month lag (in gray), and two-month lag (in dark blue) prior to the phenological event. The significance of the correlation is indicated by *p<0.01, **p<0.001, and *** p<0.0001.

Rainfall remained below 40 mm month$^{-1}$ from May to September, triggering leaf fall. Most communities presented strong negative correlations between rainfall and leaf fall intensity, with correlation coefficients above 75% for PS, IS, CS, MF, and ASDF (Fig 6). The influence of rainfall on leaf fall intensity occurred up to one month before the event in some communities, such as the IS ($r_1$ = -0.62, p = 0.02) and PS ($r_1$ = 0.63, p = 0.02) communities. Unlike the other meteorological variables, increased insolation positively influenced leaf fall phenology in open canopy communities, such as the IS ($r_0$ = 0.67, p = 0.01), PS ($r_0$ = 0.57, p = 0.04), and CS ($r_0$ = 0.56, p = 0.04) (Fig 6). Leaf fall intensity in the seasonal alluvial forest (ASDF) was also positively correlated with increased insolation in dry months ($r_0$ = 0.57, p = 0.04). There was no significant correlation between the soil moisture content and the leaf fall cycle in any of the communities, although the soil moisture showed expressive seasonal variation in most of the sampled vegetation (Fig 2).

**Flowering.** The influence of seasonality on meteorological variables and its correlation with flowering intensity varied between communities (Fig 6). In the SHB community, annual variability in soil moisture triggered flowering events, with flowering occurring in August and September, when the soil moisture dropped below 5%, and the mean (29°C) and maximum (37°C) temperatures reached their peak. The impact of evapotranspiration was observed with a two-month delay ($r_1$ = -0.62, p = 0.02). Insolation had a positive effect on flowering one month prior to the event ($r_1$ = 0.59, p = 0.03) and a negative effect in the two months preceding it ($r_2$ = -0.61, p = 0.03).

In PS and IS, an increase in insolation hours coincided with more intense flowering events (PS: $r_0$ = 0.57, p = 0.04; IS: $r_0$ = 0.56, p = 0.05). The minimum temperature had a strong influence on flowering events in PS ($r_0$ = -0.74, p = 0.004), suggesting that cold fronts can trigger flower blooming.

The cyclical change in weather significantly influenced the flowering cycles in the MF, IS, and PS communities, aligning with trends in various meteorological variables (Fig 6). For example, a reduction in evapotranspiration was associated with an increased percentage of flowering in the MF ($r_0$ = -0.86, p < 0.001), IS ($r_0$ = -0.71, p = 0.003), and PS ($r_0$ = -0.70, p = 0.01) stands. Increased insolation enhanced bloom coverage in the IS ($r_0$ = 0.56, p = 0.05) and PS ($r_0$ = 0.57, p = 0.04) communities. Additionally, reduced maximum temperatures significantly affected the flowering intensity in the two months preceding the event. In the CS, flowering intensity was weakly correlated with climate seasonality. The decrease in maximum temperature two months prior to the reproductive event was associated with increased flowering intensity ($r_2$ = -0.60, p = 0.03). Notably, the months leading to the peak of flowering were characterized by cooler temperatures of 5°C cooler. Finally, the peak flowering periods in the ASDF ($r_2$ = 0.62, p = 0.02) and LSDF ($r_2$ = 0.61, p = 0.03) forests were linked to increased rainfall in the two preceding months, marking transitions from wet to dry periods.

**Fruting.** Climate seasonality had limited influences on fruiting phenology. No correlation was found with soil moisture. Only in the SHB and MF, communities subjected to prolonged flooding (> 3 months, Fig 2), did the fruiting intensity correlate with weather variables. In the SHB, the maximum temperature ($r_0$ = 0.56, p = 0.04) during the phenological event and insolation one month prior ($r_1$ = 0.64, p = 0.02) were positively correlated with the fruiting percentage of the Fournier index, whereas insolation two months prior ($r_2$ = -0.55, p = 0.05) was negatively correlated. In the MF treatment, the mean temperature ($r_0$ = 0.57, p = 0.04) positively influenced fruiting, with higher mean temperatures two months earlier triggering high-intensity fruiting events ($r_2$ = 0.57, p = 0.04).

## Discussion

### Leaf phenology

The leaf-shedding cycles observed in the savanna and forest communities in our study ranged from weakly seasonal to aseasonal, highlighting the diverse phenological responses among woody species. This variation in leaf fall traits reflects the evolutionary and ecological forces linked to environmental filtering that enable species to thrive under different habitat conditions [74].

In all the analyzed communities, semideciduous species dominated, comprising more than 50% of the leaf exchange strategies. This aligns with Camargo et al.'s (2018) classification of the woody Cerrado as seasonal semideciduous vegetation [63]. Approximately 29% of the woody species in these communities exhibited complete deciduousness, particularly among those biogeographically associated with the Bolivian Bosque Chiquitano, mesotrophic cerradão, and other seasonal forests in South America [75]. Notable examples include *A. colubrina*, *C. fasciculata*, *H. heptaphyllus*, and *M. hirtum*. In the sandy, low-fertility soils of the PS community, the number of deciduous species doubles compared with those of the other six communities, mirroring the trend observed along the gradient from the Cerrado to seasonally dry tropical forests in northeastern Brazil, where deciduous species increase with aridity [76]. Evergreen species are present in all communities but are more abundant in seasonal seasonal forests, which are typically associated with low to moderate flooding [77].

Convergent responses to leaf shedding within communities were closely tied to climate seasonality. Increased exposure to solar radiation and reduced evapotranspiration during periods of low cloud cover positively influenced leaf fall in the dry winter, particularly in open-canopy communities such as PS and CS. Synchronizing leaf emergence and shedding with solar radiation maxima and minima enables trees to maximize water use, contributing to fluctuations in evapotranspiration rates [78]. The increase in evapotranspiration observed at the start of the rainy season corresponds with rising net radiation and temperature [79,80]. Notably, increasing maximum monthly temperatures are correlated with increased leaf fall, especially in the SHB community, which experiences several months of inundation during the wet season and very low soil moisture during the driest months [77] *C. lanceolatum*, the monodominant species in this community, behaves as an evergreen for most of the year but experiences increased leaf shedding during periods of peak temperatures. In this context, leaf shedding aids in dissipating heat during high temperatures and intense solar exposure while also reducing water loss, enhancing leaf cooling, and mitigating the risk of surpassing stress resistance thresholds [81]. Meteorological influences on the vegetative phenological cycle begin approximately two months prior to high-intensity events, underscoring the importance of predictable seasonal climate transitions—such as wet periods following droughts and the arrival of cold air masses—that trigger significant phenological shifts [82].

The climate seasonality associated with an annual flood pulse imposes additional constraints on woody plants, influencing patterns of deciduousness [83,84]. Anoxic conditions and reduced gas exchange during flooding shape various plant strategies to cope with near-zero oxygen concentrations [9]. At the end of the rainy season, when the maximum soil moisture and flood levels are reached, a second peak in leaf fall is observed, with individuals experiencing more than 50% leaf shedding across all communities. This pattern indicates that leaf shedding serves as a strategy to avoid moisture stress caused by both dry-season droughts and flood-season soil hypoxia [9,11] Understanding the relationship between vegetative phenology and water stress requires species-specific information on adaptations to drought and flood resistance [11,85]. Jancoski et al. (2022) provided evidence of interspecific variation in hydraulic function regulation influenced by deciduousness [23]. Young individuals of *T.*

*aurea*, a typical Cerrado species, tolerate flood stress by preventing root system death and leaf abscission [86]. Total or partial deciduousness, combined with strong stomatal regulation, appears to be a common water regulation strategy for both dry and flood season species in the Cerrado and flooded forests [23,81]. *L. parvifolius*, commonly found in MF, efficiently maintains water use during the dry season but exhibits stomatal limitations in response to seasonal flooding [81], with more than 50% of its leaves lost during this period. The two peaks in leaf fall intensity during both dry and wet periods have significant implications for carbon cycle dynamics within these communities, as the decomposition of senescing leaves contributes to the release of carbon from the soil back into the atmosphere [87].

Many species identified in the studied floodplain were found across multiple communities, exhibiting intraspecific variation in vegetative activity. This variation is attributed to distinct ontogenetic stages within populations, small-scale environmental heterogeneity, and genetic structure [73,88,89]. Water table depth and fluctuations are considered to modulate plasticity in maximum root depth among species occurring in both the Cerrado and Pantanal. For example, the root depth of *T. aurea* and *C. americana* in the Pantanal was found to be half that of their counterparts in the Cerrado [73]. Shallow root systems during waterlogging may lead to reduced transpiration and tree growth, triggering additional leaf-shedding events, a phenomenon also observed in Amazon floodplain trees. Some individuals of *A. discolor* were observed to partially and shortly shed leaves during the flood season, reinforcing the idea that leaf shedding can occur in evergreen species in the aquatic phase [11,83]. Phenotypic plasticity allows species populations to adapt to various positions along the soil hydrological gradient, facilitating their persistence in savannas and forests [90].

Even without clear evidence that surface soil moisture seasonality affects leaf shedding, water table dynamics may influence the ability of leaf fall to adjust during periods of low atmospheric and soil moisture. We speculate that the lack of a significant correlation between soil moisture and leaf shedding intensity may stem from habitat-specific interactions among soil physical attributes, rainfall, and flood dynamics that disrupt the alignment between rainfall timing and soil moisture seasonality [13,91]. These dynamics can lead to prolonged water shortages, reducing the seasonality of leaf shedding, as observed in Lowland Seasonally Dry Forests (LSDF). In these seasonal forests, which are typically found in areas of high soil fertility along flood-free riverbanks and relict dunes [91], the relatively high number of evergreen species suggests that some trees are buffered against seasonal drought due to access to deep soil layers (2–6 m) or storage in succulent tree trunks [92,93]. In contrast, the floristically similar ASDF may experience significant leaf fall seasonality due to stronger connections between rainfall, soil surface moisture, and groundwater fluctuations at lower elevations near the river's watercourse [91]. Access to groundwater reserves is thus a critical factor for the resilience of both forest and savanna ecosystems [73,94].

## Reproductive phenology cycle

The intricate interaction between climate signals and the reproductive patterns of plant communities in diverse ecosystems, particularly in the Pantanal region, highlights the critical role of phenological events in shaping ecological dynamics [22,95–98]. The studied communities exhibited significant seasonal flowering, with notable exceptions in the ASDF. In essence, these results from the Pantanal underscore the crucial role of lower minimum and maximum temperatures, rainfall, and evapotranspiration in triggering flowering events, highlighting drying and cooler conditions as a period of peak flower production [99]. Climatic antecedence emerges as a key determinant of the rhythms of the phenophase and is influenced by the feedback between hydroclimatic conditions and energy flux with land cover type [100].

The distinct synchronies and varied functions of plant species assemblies have led to some communities exhibiting peak flowering activity during both the dry season and wet phase, which is related to the presence of species with multiple reproductive strategies within the same community. Species associated with ASDF, such as *T. catigua* and *A. cuspa*, have individuals who flower in opposite hydrological phases in the rainy and dry seasons, respectively, which could be influenced by their distinct floristic affinities with neighboring ecosystems (e.g., Atlantic forest, Seasonal forest, and Cerrado) [75]. In general, the studied species populations exhibited low-intensity and brief-to-intermediate flowering events [101], which aligns with the findings of other studies [22,99,102,103]. Damasceno-Junior (2020) noted that the flowering phenophase in the Pantanal is generally weakly seasonal (111), gaining significance in certain species groups associated with dispersal strategies [104]. The communities with greater synchronisms were those dominated by a few species (SHB, IS, and MF) or that occurred on flood-free ground on ridges or ancient levees (i.e., LSDF) [46]. Species employing seed or fruit dispersal during periods of high river levels, such as *V. divergens* and *M. guianensis*, presented peak flowering activity at the very end of the wet season (June), when flooding receded, resulting in fruiting in January. *T. aurea* and *H. heptaphyllus*, known for their massive and synchronized flowering, produce numerous anemochoric seeds that respond clearly to variations in climate [51] The flowering phenophase in SHB occurs simultaneously with fruiting during the dry season. Synchronized reproductive behavior is influenced by high temperatures and consecutive months of exposure to solar radiation, indicating the high tolerance of savanna shrub species to extreme heat and drought [105].

In addition to climatic influence, endogenous forces play a role in explaining low and asynchronous flowering within species populations [106,107], as observed in both CS and ASDF, respectively. Endogenous factors, which are crucial for flower production, can manifest through spatial variations in the chemical attributes of the soil [106] and as a life strategy to avoid competition for visitors and pollinators [107]. Notably, the physicochemical aspects of the soil, such as the phosphorus content [108], vary among the distinct vegetation types in the Pantanal [109].

In general, the seasonal synchrony of fruiting among the different species was weak (concentration coefficient, $r_c < 0.50$) across the various vegetation communities. This phenomenon may arise when coexisting plant species exhibit diverse life cycle modes, resulting in the decoupling of phenophases from climatic seasonality [69,108,110]. At the individual level, the presence of fruits at reduced frequency and intensity (Fournier intensity < 6%) underscores the potential influence of environmentally heterogeneous ecosystems and resource availability on heightened species competition and reduced energy investment in reproduction [111,112]. This divergence in reproductive schedules establishes a resource availability timetable for fauna, promoting complementarity and potentially sustaining biodiversity throughout a longer part of the year [113].

In most of the monitored communities, species populations consistently produce fruits year-round, although at low frequency and intensity, which can be advantageous for frugivorous animals (e.g., *Tapirus terrestris* and *Ramphastos toco*), allowing them to align their fruit consumption with the phenology of various fruiting species [114–116]. Conversely, communities characterized by a strong dominance of a few species, notably SHB and MF, presented pronounced seasonality in terms of fruit production influenced by increasing temperatures during the austral wetter summer. This finding underscores the significant influence of the flowering season on subsequent fruit dispersal in these environments [104,117]. While leaf development is typically more responsive to changes in water availability than flowering or fruit production is [83], access to groundwater may allow fruiting to occur independently of the usual moisture fluctuations [118]. Additionally, species' phylogenetic signals can further decouple reproductive

cycles from water availability, leading phylogenetically related species to exhibit similar phenological periods, irrespective of moisture conditions or seasonal changes [118].

These fruiting patterns reveal a nuanced strategy mediated by dispersal syndromes and phylogenetic signals in plant phenology [22,119]. While abiotically dispersed species (mostly wind-dispersed) release seeds mainly in the dry season, the wet-season peak is driven almost entirely by biotically dispersed species [22,120]. For example, *T. catigua* and *A. discolor* tend to grow in alluvial forests and flood-prone areas [75] during the wettest months (December-January; ~24% soil moisture). On the other hand, fructification of species found in CS, such as *A. fraxinifolium* and *A. colubrina*, occurred during the driest months (August-September; < 14% soil moisture).

The relationships between plant communities and seed dispersers, which include mammals (e.g., bats and primates), birds, and fishes, underscore the critical role of flood pulses in shaping patterns of seed deposition. The intricate balance required for ecosystem health and regeneration and the role of dispersal strategies within each community assembly [108] provide valuable insights into why the onset and end of the reproductive period may not fully coincide across communities. Frugivory by fish and hydrochory are relevant mechanisms for determining lateral dispersal and seed deposition patterns across floodplain communities [121,122]. For example, *M. guianensis* and *O. suaveolens*, which undergo fruiting during the rainy season in the MF and ASDF, serve as examples of species dispersed by fish [96]. Their involvement plays a vital role in the regeneration of forest ecosystems [114].

## Concluding remarks: Informing conservation and management initiatives

The unique hydrological response of the Pantanal to the climatic cycle sets it apart from other nonflooded savanna vegetation [17,123]. This response involves the intricate interplay between phenological dynamism and alternating drought and wet phases [11,25]. This study revealed the existence of interspecific synchrony in vegetative and reproductive phenology within savanna and forest communities. Differences in seasonal phenological patterns among communities were predominantly observed in flowering, with communities with intensities concentrated from the very end of the wet season (when flooding is receding) to the dry season, differing from those with flowering events more scattered throughout the year (Figs 5A and S1). Interspecific synchronism and seasonal responses in leaf fall were triggered by climate seasonality in terms of rainfall, temperature, and evapotranspiration, while the climate signals for flowering and fruiting events were predominantly detected for communities with high dominance for few species.

The difference in assemblage activity times observed in other Pantanal regions [108] might result from the complex interplay between plant evolution and the current selective habitat filters [51,124,125], resulting in unique strategies for coping with periods of soil or atmospheric water stress concomitant with high air temperatures [126,127]. The broad array of phenological responses, which might confer stability to interannual climate fluctuations at the community level, is further supported by the ability of some species to acclimate to increasing drought or wet conditions [128] and occupy distinct zones of the soil hydrological gradient in the Pantanal [17]. Nevertheless, in the context of climate change, the northern Pantanal has experienced a shift toward aridity, marked by a 13% increase in the number of days without rain compared with that in the last 60 years. Additionally, there was a 16% reduction in water mass during the drought season compared with that observed a decade ago [129]. This approach tests tolerance limits and alters the dynamics of competitive strategies among shrubs, palms, lianas, and trees while vying for resources in challenging environments.

Through repeated cycles of drought and a warming climate, the studied area may be subjected to changes in species dominance and potential community declines due to elevated tree mortality across various ontogenetic stages [25,50]. Communities characterized by weak responses to rainfall seasonality, such as the LSDF and SHR, are anticipated to experience less impact amidst ongoing climate change. Conversely, communities that quickly adjust to yearly climate patterns may experience more significant impacts. The trajectory of these changes will depend heavily on the species-specific phenological and physiological strategies that plants adopt to cope with drier and warmer climate conditions, as well as disturbances such as fires [130]. Araújo et al. (2021) suggested that deciduousness becomes more likely in forest trees, causing a physiognomic shift toward savanna-like vegetation formation [90]. Species that exhibit leaf drop (deciduous) and employ low investment in fruits and seeds (anemochory and autochory) [108] may perpetuate this phenomenon in the context of extended and intense dry seasons, along with reduced flooding [51,88,131]. The replacement of long-lived trees with pioneer trees and lianas is also expected under such climate change scenarios and after wildfires [25,131]. Even zoochoric tree species, including the caducifolious and pioneer Cerrado species *C. americana*, may become dominant under drier climate conditions and intensified fire regimes [132]. The interactions between groundwater recharge and surface water dynamics in communities are critical in determining vegetation shifts under climate change scenarios [133]. Nevertheless, the diminished signals for flowering triggered by low temperatures may lead to decreased flowering or the prevention of flowering events in several species [106]. The potential for long-distance dispersal via zoochory and hydrochory across the floodplain is expected to be constrained, which may lead to a reduction in species richness [25] and a decrease in available resources for the diverse fauna found in the Pantanal [28,96,114].

Understanding the selective pressures that shape species' adaptations to the environment can provide insight into the mechanisms driving ecosystem change and help in conservation and management efforts [134]. Considering the substantial contribution of vertebrates to woody plant dispersal in the Pantanal, the mortality of fauna during wildfires [28] might pose critical concerns to vegetation regeneration and conservation of the diverse mosaics of plant communities in the Pantanal, reducing seedling recruitment [25,135]. The challenges posed by climate change, wildfires, and the potential loss of large-bodied vertebrates necessitate ongoing research, long-term monitoring, and conservation efforts to safeguard the unique biodiversity and ecosystem services offered by the Pantanal. Since the timing of the phenophase is irregular to some degree between years [69], long-term monitoring of phenological cycles could provide additional insight into future scenarios related to climate variability in Pantanal vegetation.

## Supporting information

**S1 Fig. Frequency of woody individuals undergoing leaf shedding, flowering, and fruiting phenological activities throughout an annual hydroclimatic cycle (July 2017 to June 2018).** The color gradient illustrates periods of high activity (intense colors) and low activity (softer colors) among individuals.
(TIF)

**S2 Fig. Frequency of woody species in reproductive activity of, A. Flowering activity and B. Fruiting activity, across the annual hydroclimatic cycle for the seven communities.**
(TIF)

**S1 Table. Woody species composition and vegetative and reproductive phenology of seven savanna and forest communities of the northern Pantanal.** A total of 59 species from 28 families were recorded, totaling 193 individuals in 7,200 m$^2$. The table presents the occupied

communities and the months of the year (from 1 to 12, corresponding to January to December) for the activities of leaf fall, flowering, and fruiting of each species. Leaf fall activity is considered when it is above or equal to 50%. PL < 50%: Partial Leaf Fall (less than 50%); LY: Leaf Fall Lost Year-round ($\geq$ 50%). Communities: Cerrado Savanna (CS), Invaded Savanna (IS), Parkland Savanna (PS), Shrubland (SHB), Lowland Seasonal Semideciduous Forest (LSDF), Alluvial Seasonal Semideciduous Forest (ASDF), and Semi-evergreen Monodominant Forest of *Vochysia divergens* (MF).
(DOCX)

**S1 Appendix. Dataset containing the original data utilized in the phenological study conducted in the Pantanal Mato-grossense, SESC Baía das Pedras, Poconé municipality, Mato Grosso (Brazil).** It encompasses information on reproductive and vegetative phenology, environmental data, and results from derived analyses.
(XLSX)

## Acknowledgments

We dedicate this work to the memory of the UFMT technician Hélio Ferreira for his invaluable support, which contributed significantly to our research and that of many young scientists in plant ecology at the UFMT. We are thankful for the technical and logistic support provided by the Federal University of Mato Grosso, the SESC Pantanal (Brazilian Social Service of Commerce), the Pantanal Research Center (CPP), and the National Institute for Science and Technology of Wetlands (INAU).

## Author Contributions

**Conceptualization:** Julia Arieira, Karl-L. Schuchmann, Marinêz Isaac Marques.

**Data curation:** Julia Arieira.

**Formal analysis:** Julia Arieira.

**Funding acquisition:** Karl-L. Schuchmann, Marinêz Isaac Marques.

**Investigation:** Julia Arieira, Michelle D. Lanssanova.

**Methodology:** Julia Arieira.

**Project administration:** Julia Arieira, Karl-L. Schuchmann, Marinêz Isaac Marques.

**Resources:** Julia Arieira, Karl-L. Schuchmann, Marinêz Isaac Marques.

**Software:** Julia Arieira.

**Supervision:** Julia Arieira, Karl-L. Schuchmann, Marinêz Isaac Marques.

**Validation:** Julia Arieira.

**Visualization:** Julia Arieira, Ana Silvia O. Tissiani, Osvaldo Borges Pinto Junior.

**Writing – original draft:** Julia Arieira, Michelle D. Lanssanova.

**Writing – review & editing:** Julia Arieira, Karl-L. Schuchmann, Arnildo Pott, Osvaldo Borges Pinto Junior, Marinêz Isaac Marques.

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
