## [Decision Letter · Decision Letter 0]

5 Jul 2024

PONE-D-24-13579Phenological patterns and climate influences on vegetation communities in the northern PantanalPLOS ONE

Dear Dr. Arieira,

Thank you for submitting your manuscript to PLOS ONE. After careful consideration, we feel that it has merit but does not fully meet PLOS ONE’s publication criteria as it currently stands. Therefore, we invite you to submit a revised version of the manuscript that addresses the points raised during the review process.

Two experts have recommended accepting your manuscript for publication. Please note that one of the reviewers raised relevant concerns regarding the lack of detail in the Methods section and other specific comments. I agree with his assessment and recommend some additional changes to improve the presentation of your manuscript, as indicated below. I am willing to consider a revised version for publication in this journal, assuming you modify the manuscript according to all recommendations. 

We look forward to receiving your revised manuscript.

Kind regards,

Angelina Martínez-Yrízar, Ph.D.

Academic Editor

PLOS ONE

Journal Requirements:

https://journals.plos.org/plosone/s/file?id=ba62/PLOSOne_formatting_sample_title_authors_affiliations.pdf"

3. We note that your Data Availability Statement is currently as follows: "All relevant data are within the manuscript and its Supporting Information files."

5. We notice that your supplementary table are included in the manuscript file. Please remove them and upload them with the file type 'Supporting Information'. Please ensure that each Supporting Information file has a legend listed in the manuscript after the references list.

6. Please upload a copy of Supporting Information Figure/Table/etc. Table which you refer to in your text on page 48.

Additional Editor Comments:

I find the title unclear, not accurate: “Phenological patterns and climate influences on vegetation communities in the northern Pantanal”…phenological patterns influence vegetation communities? My suggestion is to draft the title correctly, aligned with the aims of your study (lines 27-29 and lines 99-101).

Line 73, “Pantanal mato-grossense” For clarity to readers, provide in brackets a definition of this Pantanal type.

Line 145, indicate what is the total extension of your study area.

Line 149-150, make clear that you are referring here to species numbers.

Line 153, “invaded by trees” exotic or native? Make this point clear.

Line 245, as I understand, all your climatic analysis is based on data provided by a single meteorological station, Cuiabá, Mato Grosso (OMM: 83361). Are these data representative of the climate for the whole area? Explain.

Lines 261-264, indicate how these temperature and precipitation values differ/depart from long-term (30 yrs or available historical data) mean values.

Lines 166-167, “made in 21 plots (one to three per community) measuring 5 m x 80 m, totaling a sampled area of 8.84 ha”. This explanation is not clear to me. If plots were 1-3 per community and you sampled 7 communities, then how a total of 21 plots Also, 5 m x 80 m size plot x 21 plots = 8,400 m2 or 0.84 ha, how 8.84 ha? Rewrite.

Line 172, explain the criterion employed for selecting the 982 individuals and specify the number of families to which those species belong.

Line 295, this is the first time you refer to the “hydrological gradient” and to the extremes of that gradient (spatial or temporal?). These must be explicitly described in the Methods section. Also, seems to me you are using the following terms indistinctly: flooding regime, hydrological gradient, hydrogeomorphological gradients, and flooding regime gradients. Please correct.

Line 644, add the reference number to the Lima and Damascenso-Junior (2020) citation.

Reviewers' comments:

Reviewer's Responses to Questions

**Comments to the Author**

1. Is the manuscript technically sound, and do the data support the conclusions?

Reviewer #1: Yes

Reviewer #2: Partly

2. Has the statistical analysis been performed appropriately and rigorously? 

Reviewer #1: Yes

Reviewer #2: Yes

3. Have the authors made all data underlying the findings in their manuscript fully available?

Reviewer #1: Yes

Reviewer #2: Yes

4. Is the manuscript presented in an intelligible fashion and written in standard English?

Reviewer #1: Yes

Reviewer #2: Yes

5. Review Comments to the Author

Reviewer #1: No comments to the authors

Reviewer #2: The writing of the paper lacks clarity in presenting the seven types of vegetation communities. They are mentioned only between lines 152 and 158, without detailing the environmental or hydrogeomorphological differences that characterize each community. Furthermore, it is not clear whether the level of conservation or degradation is similar among them. The absence of this information makes it difficult to relate and analyse the results.

This information should be clearly presented in the "Sampling Design and Collection" section, and the results should be organized based on these differentiations.

Another fundamental point is the lack of clarity in the "Concluding Remarks" section. It is not clear whether there is synchronization or not in the phenology of the different communities and what the contribution of environmental seasonality is to this.

Other points:

In the results on soil moisture, does the result of 100% mean that the ground was flooded? If so, it needs to be made clear.

In the section "Influence of Climate and Soil Moisture on Fournier's Percent Index," there is no presentation or discussion of the flowering data. Was there no influence?

Line 576: Are any of the analysed communities classified as nonflooded Cerrado?

Table 02: What do the variations of blue and red colours represent?

6. PLOS authors have the option to publish the peer review history of their article (what does this mean?). If published, this will include your full peer review and any attached files.

Reviewer #1: **Yes: **Cleber J. R. Alho

Reviewer #2: No

---

## [Author Response · Author response to Decision Letter 0]

26 Nov 2024

PLOS ONE – 'Response to Reviewers'

PONE-D-24-13579

Phenological patterns and climate influences on vegetation communities in the northern Pantanal

PLOS ONE

Dear Dr. Angelina Martínez-Yrízar,

Thank you for considering our manuscript for publication in PLOS ONE. After thoroughly reviewing the feedback provided by you and the reviewers, we are pleased to submit the revised version of the manuscript.

We took this opportunity to carefully recheck the data and analyses and have incorporated new content, making revisions to address the reviewers' suggestions. Additionally, we have ensured that the manuscript adheres to PLOS ONE's style guidelines.

Enclosed, you will find our detailed responses to the comments raised by both you and the reviewers.

We hope that the revisions meet your expectations and have enhanced the quality of the manuscript. We remain enthusiastic about the opportunity to publish with PLOS ONE.

Kind regards,

Julia Arieira (corresponding author)

Response to Reviewers and Editor

https://journals.plos.org/plosone/s/file?id=ba62/PLOSOne_formatting_sample_title_authors_affiliations.pdf"

Authors’ Response: We have reviewed the style requirements and made the necessary edits.

Authors’ Response: We included the following sentence about permission for the work: ‘Since the research was conducted outside protected areas and did not involve endangered species, no data collection permission from official environmental institutions (i.e., Instituto Chico Mendes de Conservação da Biodiversidade - ICMBio) was needed.’

3. We note that your Data Availability Statement is currently as follows: "All relevant data are within the manuscript and its Supporting Information files."

Authors’ Response: We are uploading a spreadsheet (Appendix S1) with data used for analyses and produce tables and figures as supporting information.

4. We note that Figure 1 in your submission contains [map/satellite] imagthat may be copyrighted. All PLOS content is published under the Creative Commons Attribution License (CC BY 4.0), which means that the manuscript, images, and Supporting Information files will be freely available online, and any third party is permitted to access, download, copy, distribute, and use these materials in any way, even commercially, with proper attribution. For these reasons, we cannot publish previously copyrighted maps or satellite images created using proprietary data, such as Google software (Google Maps, Street View, and Earth). For more information, see our copyright guidelines: http://journals.plos.org/plosone/s/licenses-and-copyright.

Authors’ Response: We replaced the figure.

5. We notice that your supplementary table are included in the manuscript file. Please remove them and upload them with the file type 'Supporting Information'. Please ensure that each Supporting Information file has a legend listed in the manuscript after the references list.

Authors’ Response: We removed the table within the manuscript and are submitting it as Table S1 Table. We also included legends after references as requested.

6. Please upload a copy of Supporting Information Figure/Table/etc. Table which you refer to in your text on page 48.

Authors’ Response: Supporting information is provided in the Appendix.

7. Additional Editor Comments:

I find the title unclear, not accurate: “Phenological patterns and climate influences on vegetation communities in the northern Pantanal”…phenological patterns influence vegetation communities? My suggestion is to draft the title correctly, aligned with the aims of your study (lines 27-29 and lines 99-101).

Authors’ Response: We replaced the title with: “Phenological Cycles in Pantanal Woody Communities: Responses to Climate and Soil Moisture Seasonality”

8. Line 73, “Pantanal mato-grossense” For clarity to readers, provide in brackets a definition of this Pantanal type.

Authors’ Response: We added the classification of Pantanal wetland by Junk et al. (2015): ‘a floodplain wetland subjected to predictable monomodal flood pulse of low amplitude’

9. Line 145, indicate what is the total extension of your study area.

Authors’ Response: We included the total extension of the study area and improved the whole paragraph.

10. Line 149-150, make clear that you are referring here to species numbers.

Authors’ Response: Review.

11. Line 153, “invaded by trees” exotic or native? Make this point clear.

Authors’ Response: Invaded was replaced by ‘invaded by pioneer and native trees’. These species encroach savanna areas becoming a socioeconomic issue for cattle ranchers in the Pantanal; thus, they are called invasive species by farmer. We included a brief description of the seven communities to provide a better understanding of the results. 

12. Line 245, as I understand, all your climatic analysis is based on data provided by a single meteorological station, Cuiabá, Mato Grosso (OMM: 83361). Are these data representative of the climate for the whole area? Explain.

Authors’ Response: Yes, it is. This station has been the official weather station in operation since 1911 (https://portal.inmet.gov.br/normais.). It is used to calculate the climatological normals of Brazil and serves as a reference for climate monitoring in the North Pantanal. We have included this information in the updated version of the manuscript.

The graphic below shows a comparison between the annual seasonality in precipitation between Cuiabá and Poconé.

13. Lines 261-264, indicate how these temperature and precipitation values differ/depart from long-term (30 years or available historical data) mean values.

Authors’ Response: To address this comment, we included additional information: 

‘This study was performed during the annual hydroclimatic cycle from July 2017 to July 2018 in the northern Pantanal (16°30’S and 56º25’W) (Fig. 1). This period represents normal to moderately wet conditions intercepted with intense rainy periods lasting from nearly 2-4 months [43], considering that the cumulative annual average precipitation in the region was 1,516 mm from 1991-2020 [44].

14. Lines 166-167, “made in 21 plots (one to three per community) measuring 5 m x 80 m, totaling a sampled area of 8.84 ha”. This explanation is not clear to me. If plots were 1-3 per community and you sampled 7 communities, then how a total of 21 plots Also, 5 m x 80 m size plot x 21 plots = 8,400 m2 or 0.84 ha, how 8.84 ha? Rewrite.

Authors’ Response: Thank you for warning us about this mistake. The sentence was reviewed.

15. Line 172, explain the criterion employed for selecting the 982 individuals and specifies the number of families to which those species belong.

Authors’ Response: Individuals were selected on the basis of their occurrence within the plots. We have rewritten the paragraph to clarify the methodology clearer including the criterion and number of families. 

16. Line 295, this is the first time you refer to the “hydrological gradient” and to the extremes of that gradient (spatial or temporal?). These must be explicitly described in the Methods section. Also, seems to me you are using the following terms indistinctly: flooding regime, hydrological gradient, hydrogeomorphological gradients, and flooding regime gradients. Please correct.

Authors’ Response: We have unified the terms with respect to the soil hydrological gradient (e.g., soil moisture and flood regime). 

17. Line 644, add the reference number to the Lima and Damasceno-Junior (2020) citation.

Authors’ Response: Included

18. Reviewer #2: The writing of the paper lacks clarity in presenting the seven types of vegetation communities. They are mentioned only between lines 152 and 158, without detailing the environmental or hydrogeomorphological differences that characterize each community. Furthermore, it is not clear whether the level of conservation or degradation is similar among them. The absence of this information makes it difficult to relate and analyse the results.

This information should be clearly presented in the "Sampling Design and Collection" section, and the results should be organized based on these differentiations.

Authors’ Response: We have included a description of the structural and floristic characteristics of the seven types of vegetation communities and of the habitats to which they are usually associated. We also mentioned that the study area is part of the SESC Pantanal Ecological Station, located in the municipality of Poconé (Mato Grosso, Brazil), and it is dedicated to ecotourism and scientific research.

19. Another fundamental point is the lack of clarity in the "Concluding Remarks" section. It is not clear whether there is synchronization or not in the phenology of the different communities and what the contribution of environmental seasonality is to this.

Other points:

Authors’ Response: We addressed this concern, including the following text:

“This study revealed the existence of interspecific synchrony in vegetative and reproductive phenology within savanna and forest communities. Differences in seasonal phenological patterns among communities were predominantly observed in flowering, with communities with intensities concentrated from the very end of the wet season to the dry season, differing from those with flowering more evenly distributed throughout the year. Interspecific synchronism and seasonal responses in leaf fall were triggered by climate seasonality in terms of rainfall, temperature, and evapotranspiration, while the climate signals for flowering and fruiting events were predominantly detected for communities with high dominance for few species.“

20. In the results on soil moisture, does the result of 100% mean that the ground w: as flooded? If so, it needs to be made clear.

Authors’ Response: In the ‘Abiotic control of phenological events’ section, we indicated that ‘A soil moisture content of 100% indicates that the site is in a flooded state.’

21. In the section "Influence of Climate and Soil Moisture on Fournier's Percent Index," there is no presentation or discussion of the flowering data. Was there no influence?

Authors’ Response: Thank you for waring that. We have missed something during our internal review process. We have returned to the paragraph about flowering. 

22. Line 576: Are any of the analysed communities classified as nonflooded Cerrado?

Authors’ Response: No. We have rewritten the sentence to make this clearer. We meant ‘nonflooded Cerrado which border the Pantanal’. The text was reviewed to make this clear.

23. Table 02: What do the variations of blue and red colours represent?

Authors’ Response: This table was transformed into a figure as supplementary material (Figure S1). The figure shows the distribution of individuals with vegetative and reproductive phenological activity across the annual hydroclimatic cycle (2017-2018) in seven savanna and forest communities found in a floodplain wetland in the northern Pantanal. The color gradient represents a low to high frequency of activity among individuals.

---

## [Editor Report · Decision Letter 1]

5 Dec 2024

Phenological cycles in the Pantanal woody communities: Responses to climate and soil moisture seasonality

PONE-D-24-13579R1

Dear Dr. Arieira,

We’re pleased to inform you that your manuscript has been judged scientifically suitable for publication and will be formally accepted for publication once it meets all outstanding technical requirements.

Kind regards,

Angelina Martínez-Yrízar, Ph.D.

Academic Editor

PLOS ONE

Additional Editor Comments (optional):

Please correct:

Pag 4. Revise this assertion “under a scenario of 30% rainfall reduction and a 7ºC temperature increase by 2100”. 7 oC? According to IPCC: the global temperature could rise by 3.3°C to 5.7°C (5.9°F to 10.3°F) by 2100 in a high-emission scenario. And provide a reference.

Figure 4. Move the word “leaf fall” to the upper part of the graphs. It is located out of place.
---

## [Editor Report · Acceptance letter]

20 Dec 2024

PONE-D-24-13579R1 

PLOS ONE

Dear Dr. Arieira, 

I'm pleased to inform you that your manuscript has been deemed suitable for publication in PLOS ONE. Congratulations! Your manuscript is now being handed over to our production team.

Kind regards, 

on behalf of

Dr. Angelina Martínez-Yrízar 

Academic Editor

PLOS ONE
